# Alternative method to visualize receptor dynamics in cell membranes

**Ravelli Cosetta**[1,2]*, **Corsini Michela**[1,2], **Ventura Anna**[1], **Domenichini Mattia**[1], **Grillo Elisabetta**[1,2], **Mitola Stefania**[1,2]*

1 Department of Molecular and Translational Medicine, University of Brescia, Brescia, Italy, 2 The Mechanobiology Research Center, UNIBS, Brescia, Italy

* cosetta.ravelli@unibs.it (RC); stefania.mitola@unibs.it (MS)

## Abstract

There is a close relation between membrane receptor dynamics and their behavior. Several microscopy techniques have been developed to study protein dynamics in live cells such as the Fluorescence Recovery After Photobleaching (FRAP) or the Single Particle Tracking (SPT). These methodologies require expensive instruments, are time consuming, allow the analysis of small portion of the cell or an extremely small number of receptors at a time. Here we propose a time-saving approach that allows to visualize the entire receptor pool and its localization in time. This protocol requires an epifluorescence microscope equipped for structured illuminated sectioning and for live cell imaging. It can be applied to characterize membrane receptor and multi protein complex and their response to activators or inhibitors. Image acquisition and analysis can be performed in two days, while cells and substratum preparation require a few minutes a day for three days.

## Introduction

Cell surface transmembrane receptors transduce extracellular cues into intracellular signaling. These proteins can be either associated to or integrated into the cell membrane. Membrane receptors are mainly divided into 3 classes: the ion channel receptors which are multi-pass transmembrane proteins, the G protein anchored receptors with seven transmembrane helices and the enzyme-linked receptors with a single-pass transmembrane helix. Among the later the Tyrosine Kinase Receptors (RTKs) play important roles in the regulation of cell growth, differentiation, and survival. They bind and respond to growth factors and other locally released proteins that are present at low concentrations. Upon the interaction with their specific ligand, RTKs dimerize with neighboring receptors and trans-phosphorylate their cytoplasmic tails which serve as docking sites for various intracellular proteins involved in signal transduction. Although several RTK ligands are soluble molecules, in tissues these are often associated with different extracellular matrix (ECM) components, including heparan-sulfate proteoglycans (HSPGs), leading to the formation of immobilized ECM-bound complexes. Several evidence shows that angiogenic growth factors, including VEGFs, can be found associated with endothelial ECM *in vitro* and with blood vessel basement membranes *in vivo* [1, 2].

The plasma membrane closed in contact with ECM was formally considered a 2D solution in which proteins and lipoproteins represented the solute dissolved in a viscous solvent

**Data Availability Statement:** All relevant data are within the manuscript and its Supporting Information files.

**Funding:** This work was supported by grants from Associazione Italiana per la Ricerca sul Cancro

(AIRC) to S.M. (IG 2021 Id25726), from MIUR to Consorzio Interuniversitario di Biotecnologie (CIB) to C.R., E.G., and S.M.; from "PNRR M4C2-Investimento 1.4-CN00000041 finanziato dall'Unione Europea–NextGenerationEU" to S.M and M.C. "The Mechanobiology Research Center" was supported by liberal donations of Copan and Antares Vision. The funders had no role in study design, data collection and analysis, decision to publish, or preparation of the manuscript.

**Competing interests:** The authors have declared that no competing interests exist.

consisting of phospholipids forming the so called "fluid mosaic" [3]. Thus, plasma membrane can be considered as a semipermeable barrier at the interface with tissue microenvironment. Membrane has a thickness of 5–10 nm consisting of a phospholipid bilayer enriched with lipids, proteins, and sugars. Most of the membrane activities are carried out by membrane proteins which can be integral, peripheral, or surface associated. In contrast with the initial assumption, it is well known now that the plasma membranes do not have a homogeneous composition and that proteins, as well as lipids, tend to concentrate in particular areas, giving rise to different membrane domains that perform different biological activities, e.g. lipid rafts, caveolae and focal adhesions [4]. Membrane proteins and lipids, as well as cortical actin (the cytoskeleton in direct contact with the plasma membrane) influence the formation and the remodeling of these membrane domains, finally defining their function and cell response to external stimuli [5]. These membrane structures govern the dynamics of proteins in the plasma membrane and their availability for ligands, co-receptors, and intracellular scaffolds. We can described the lateral regulation of RTK dynamics as a fine regulatory ways to modulate the cell/environment interactions [6, 7]. The relocation of RTKs in membrane domains is facilitated by the membrane fluidity. For example, in endothelium the spatial regulation of RTK also influences the fate and polarity of cells. Indeed, receptor localization defines the apical, the apical-lateral and the basal portion of the plasma membrane and their functions. The cell basal portion has been mainly studied for its interactions with the basal lamina, the apical lateral modulates the intravasation and extravasation fluxes and the apical portion is exposed to blood flow. In blood vessels, for example, the podocalyxin, VE-Cadherin and β1 integrin respectively characterize the different cell portions while the localization of other proteins (e.g. growth factor receptors) are non-restricted and they freely relocalize regulating different biological aspects [2, 8, 9].

The Vascular Endothelial Growth Factor Receptor 2 (VEGFR2) is recruited at the leading edge in growing blood vessels while it relocalizes in the cell junction just in contact with VE-Cadherin or at the basal contact. We recently showed that VEGF- or gremlin-enriched matrix triggers a rapid relocation and long-term activation of VEGFR2 at the basal aspect of motile endothelium [2, 10]. The different receptor localization also determines its bioavailability for extracellular ligands or drugs. It therefore becomes important to be able to analyze the dynamic capacity of the receptors and their localization. The analysis of receptor dynamics in live cells can be achieved through different microscopy-based assays, including Fluorescence recovery after photobleaching (FRAP) [10, 11] and Single particle tracking (SPT) [12, 13], that allow following respectively the collective or single molecule dynamics of RTKs. These assays need a Laser Scanning Confocal (LSC) or a Total Internal Reflection Fluorescence (TIRF) microscopy, to photo-bleach a membrane area or to analyze only the portion of membrane in close contact with the coverglass, respectively.

Here we propose an economic and easy assay to follow and characterize the membrane receptor dynamics using an epifluorescent microscopy (Fig 1). Briefly, fluorescent–tagged receptor expressing cells were cultured on coverglass and, at the time of observation, coverglass was flipped on ligand-adsorbed coverglass mimicking the first step of migration process (Fig 2). Under this experimental condition, receptors are recruited in the area close in contact with their ligands while cells progressively attach to the new substrate. Similar to migrating cell, adherent cell tests the microenvironment, reorganize the adhesion contacts, move to the new substrate and detach from the previous sites.

## Materials and methods

Step by step protocol is included in Protocol.io (*DOI: dx.doi.org/10.17504/protocols.io. rm7vzj8z8lx1/v1*).

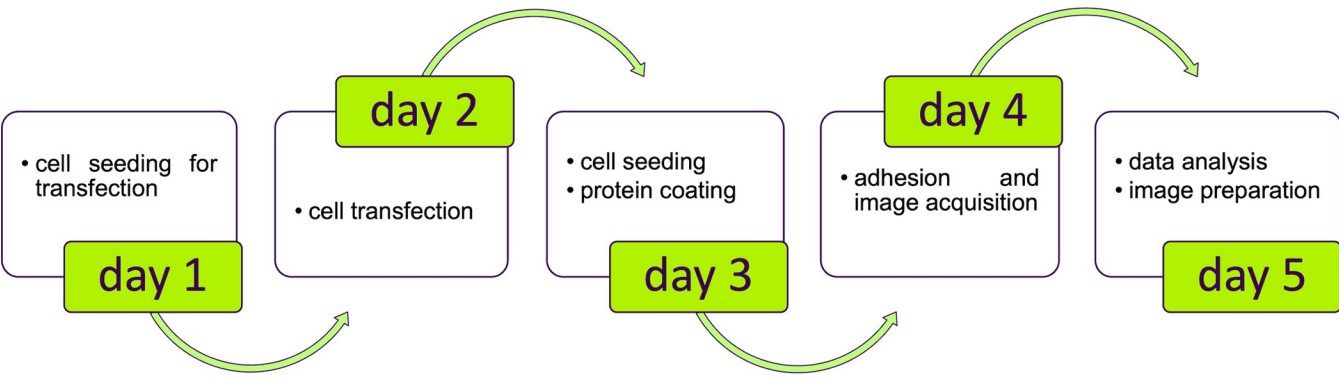

**Fig 1. Experimental flow chart.**

The primary aim of this study was to establish a new, cheap, and easy method to study receptor dynamics on plasma membranes. Unlike other methods, this protocol allows to follow in time the reorganization of the receptor using an epifluorescence microscope equipped with a structured illumination system. The experiments are conducted with cells stably expressing receptors tagged with fluorescent proteins. The transient transfection of cells allows, in the case of VEGFR2, to increase the receptor expression exposed on the plasma membrane.

## Materials

1. Pre-cleaned round borosilicate coverslips (12 mm diameter; 0.1 mm thickness) (Menzel Glaser, #CB00120RA120).

2. 6 well and 24 well cell culture plate (Corning, #3516).

3. Needle.

4. Tweezers.

5. 2-wells chambered coverslips for use in immunofluorescence and high-end microscopy (0.15mm thickness) (IBIDI, #80286).

6. Chinese Hamster Ovary (CHO) cells (ATCC, https://web.expasy.org/cellosaurus/CVCL_0213) or other adherent cells.

7. Polyethylenimine (PEI) (MERCK, 764965).

8. HAM'sF12 medium (Gibco, #11550043), eventually supplemented with (10U/mL) Penicillin-streptomycin (Gibco, #15140122) and 10% or 1% fetal calf serum (FCS).

9. pBE-hVEGFR2-eYFP, kindly provided by Dr. Kalina Hristova, Johns Hopkins University, Baltimore, USA.

10. PBS for cell culture (Gibco, #10010023).

11. Bovine Serum Albumin (BSA) (Thermofisher #15561020).

12. Recombinant VEGF-A165 (R&D system, #293-VE-010/CF).

## Equipment

• Cell culture incubator.

- Epifluorescence microscope equipped with structured illumination system for optical sectioning, a high magnification and high-resolution objective, incubation system and a monochrome camera (Axio Observer equipped with Apotome 3, Axiocam 305 mono and Plan-Apochromat 63X/1.4 Oil; Zeiss Italia SPA).

- Fiji software (https://imagej.net/software/fiji/).

## Expected results

Here we used the vascular endothelial growth factor receptor 2 (VEGFR2), a typical tyrosine kinase receptor, to set up a method to easily follow and quantify the relocation of the membrane receptor upon extracellular ligand stimulation using epifluorescence microscopy paired with structured illumination sectioning. As extracellular ligand and stimulus we used VEGF-A 165 which binds VEGFR2 with high affinity. Although the VEGF family members are considered soluble molecules, many isoforms contain heparin binding domains and in tissues are therefore bound to the heparan sulfates of the extracellular matrix (ECM). So, to mimic what happens in vivo we followed the dynamics of VEGFR2 in pre-seeded CHO cells stimulated with immobilized VEGF-A 165. Cells transfected with a pBE plasmid encoding hVEG-FR2-EYFP, were completely attached before their interaction with VEGF-A-enriched ECM. Indeed, cells were cultured for 24 hours on coverslip in complete growth medium. Under this culture condition, VEGFR2 was positioned in its physiological location maintaining its dynamics. The coverslip with cells were placed, face down, on a chamber slide pre-coated with VEGF-A 165 and incubated in an incubator on the microscope (Fig 2).

A hair was used as a spacer between the two glasses to avoid cell crushing. The cells quickly reacted to the stimulus and polarized, directing the receptor towards the ligands and attached to it. The reorganization of VEGFR2 on the cell membrane was analyzed by acquiring Z-stack images of the cell at different times (from 5 to 120 minutes). Orthogonal projection showed that all VEGFR2-EYFP were expressed on cell membranes (Fig 3A and *dx.doi.org/10.17504/protocols.io.rm7vzj8z8lx1/v1*).

The time series of orthogonal projections showed the evolution over time of the reorganization of VEGFR2 on the cell that, sensing the stimulus of VEGF-A, organized itself to adhere to the new substrate. The Z-stack reconstruction clearly showed that VEGFR2 moved towards the immobilized VEGF-A until the cell completely adhered to the stimulus. At 120 minutes, the cells completely detached from the slide to which they were previously attached to move towards the stimulus (Fig 3A). To calculate the percentage of receptor recruited in close contact to growth factors, fluorescence-positive areas of each Z-stack section were quantified

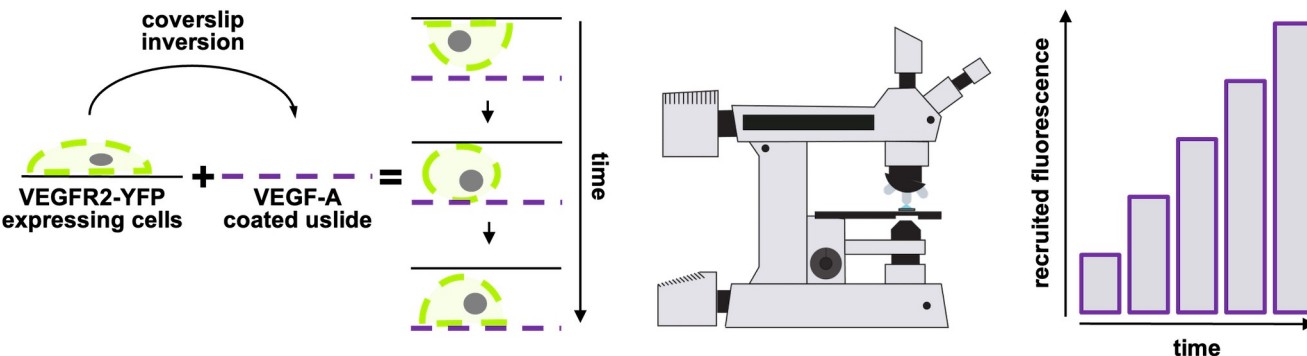

**Fig 2. Cartoon of steps 3–5 of experimental flow chat.**

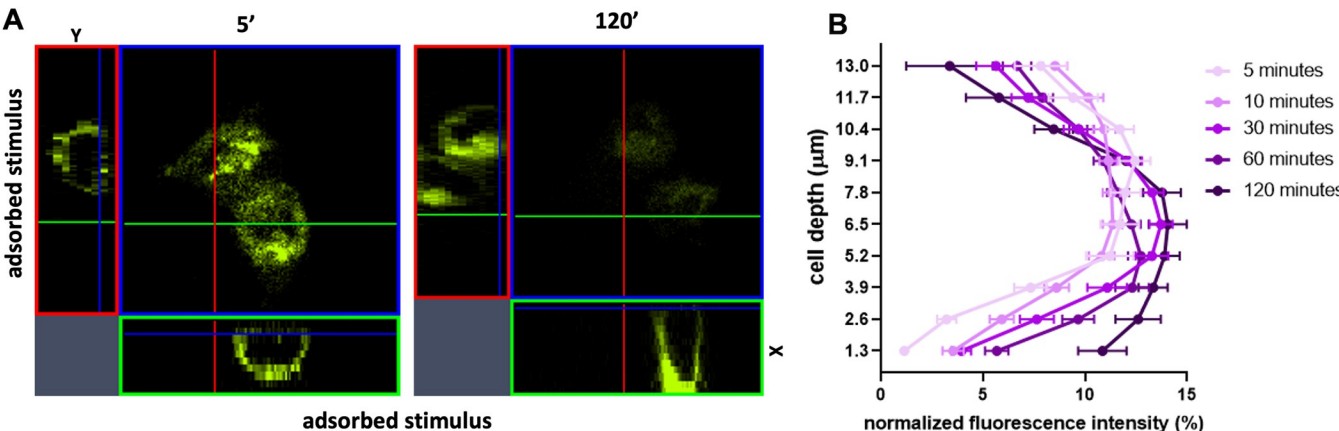

**Fig 3. Orthogonal projection of VEGFR2-EYFP expressing CHO cells adhering to VEGF-A and fluorescence quantification.** (A) Image of the apical stack and the X and Y projection of total Z-stack of cells 5 and 120 minutes after ligand interaction. (B) Darkening purple curves represent the distribution of normalized fluorescence intensity (%) in stacks during time (5, 10, 30, 60 and 120 minutes). Images are representative of 3 independent experiments that gave similar results. Data were obtained from 20 cells for each time point in 3 independent experiments.

using Fiji following instructions detailed in step 5 of the step-by-step protocol described in (*DOI: dx.doi.org/10.17504/protocols.io.rm7vzj8z8lx1/v1*). By analyzing single stacks with Fiji, distribution curves of VEGFR2 at different times were also constructed. Fig 3B shows the percentage of VEGFR2 in Z stacks starting from 5 up to 120 minutes. The passage of time is represented by the darkening of the purple curves. At 5 minutes, the receptor appears to be uniformly distributed, with a small amount in the apical portion of the cell, due to the shape of the cell. It is possible to appreciate a progressive bending of the curves towards the first Z stack which is the closest to VEGF. This demonstrates that VEGFR2 is able to polarize when stimulated by an immobilized ligand. The increasing amount of VEGFR2-EYFP in close contact with immobilized VEGF can be visualized using the 3D reconstruction with depth coding option. The entire thickness of 13 μm was represented with a color coding in which blue is the starting coverslip and red the immobilized VEGF. The red color increased with the adhesion and after 120 minutes of adhesion red color was preponderant compared to blue (Fig 4A). The recruited receptor was quantified in stacks close to VEGF (Fig 4B).

This assay represents an easy method to analyze the relocation and protein diffusion on cell membranes. This represents an alternative method to FRAP and to SPT. All of these are fluorescent-based methods, but they need respectively an epifluorescence, a confocal and a total internal reflection fluorescence (TIRF) microscope. Similar to the method described above FRAP, using a laser scan, measures the average diffusion characteristics of molecules. Both techniques are an averaging method, i.e. the data derive from the average behavior of several fluorescent molecules over time and space. Although FRAP is a sensitive method, it is not free from important technical limits: first, the high intensity lighting used to extinguish the fluorescence can induce photo-damaging effects of the cell. This can be overcome using epifluorescence microscopy in which the lamp is less energetic than a laser. To finely characterize the trajectory of the molecule the SPT is a very useful method. Although SPT gives more information, it is an extremely time-consuming technique (Table 1).

Of note, the acquisition and image reconstruction can also be performed with a laser scan confocal microscope, even if more time consuming. Acquiring images with a camera allows the analysis of more than one cell at a time in Z-stack and time lapse. Also, at the end of analysis cells can be fixed and immunostained. The increased membrane area and its close contact with coverslip facilitates the analysis and the quantification of the positive area [14, 15] and

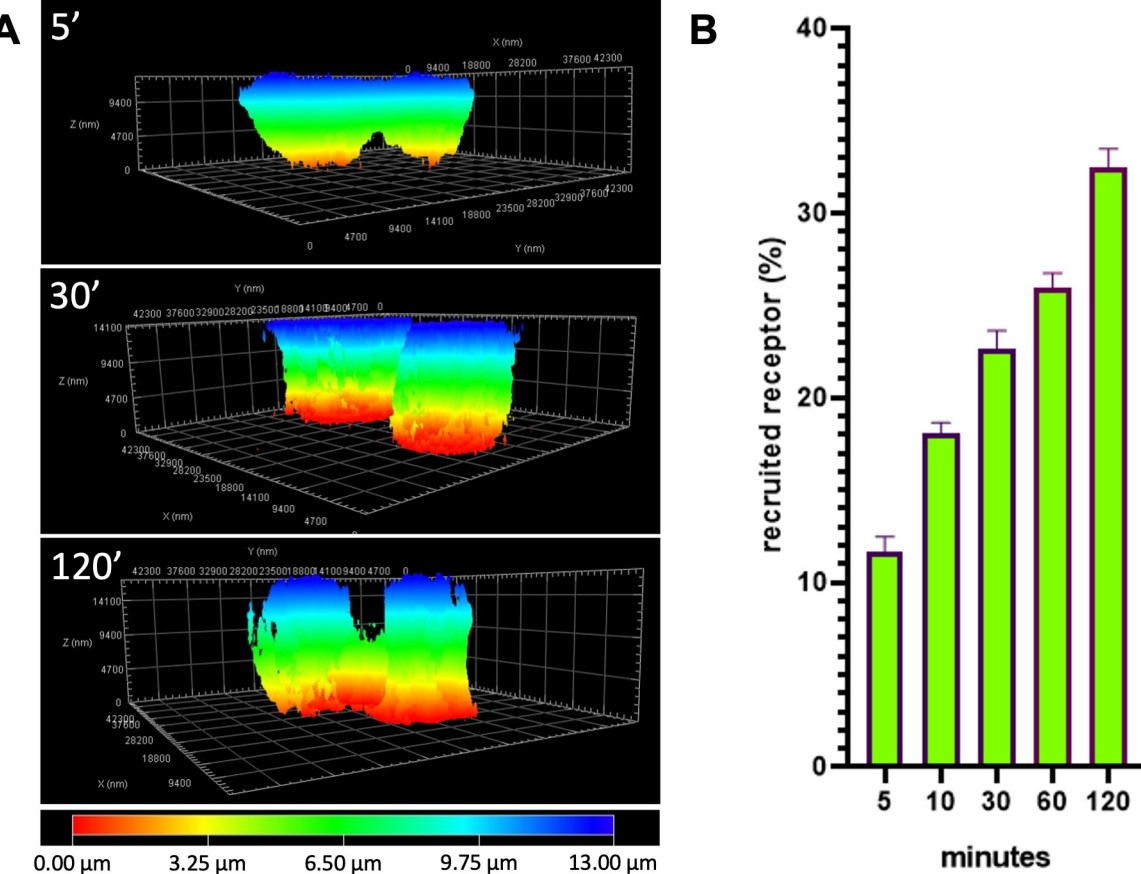

**Fig 4. 3D reconstruction of EYFP-VEGFR2 recruitment in CHO.** (A) Depth coding of 3D reconstruction of EYFP-VEGFR2. Red color represents the portion of the cell in close contact with immobilized VEGF while blue color represents the portion of the cell detaching from the coverslip. (B) Quantification in time of VEGFR2 recruitment in the basal portion of the cell. Images are representative of 3 independent experiments that gave similar results. Data were obtained from 20 cells for each time point in 3 independent experiments.

eventually allows to perform FRET analysis. In FRAP experiments the receptor dynamics can be observed in all membrane areas which are extremely plastic upon stimulation, while in SPT the time of analysis is extremely shorter.

We believe that this method is easily achievable in many basic laboratories and can provide guidance in a relatively short time. The recruitment assay will be useful to characterize not

**Table 1. Advantages and disadvantages of microscopy techniques for the study of molecule dynamics.**

| Microscopy techniques | Advantages | Disadvantages |
|---|---|---|
| **Whole cell 4D fluorescence reconstruction** | Cheap; | Bulk visualization; |
| | Whole-cell visualization; | Need of structured illumination microscopy or image deconvolution. |
| | Chance to follow receptor internalization. | |
| **Fluorescence Recovery After Photobleaching (FRAP)** | Fast identification of mobile and immobile molecular fractions; | Characterization of a restricted area of the cell; |
| | Easy definition of Diffusion Coefficient. | Photodamage; |
| | | Need of Confocal Laser Scanning Microscopy (CLSM). |
| **Single Particle Tracking (SPT)** | Visualize the position of single molecule and record its trajectories; | Need of Total Internal Reflection Fluorescence (TIRF) microscopy; |
| | High temporal and spatial resolution | Large amount of data to analyze. |

only the ligand-receptor interactions increasing the area of observation and make easier the study the direct interaction but also can be used to visualize the formation of multi receptors complex in cell membrane or to the recruitment of intracellular mediators. We showed that the recruitment of VEGFR2 by gremlin is followed by the recruitment of β3 integrin, which forms an active complex with VEGFR2, and of PI 3-kinase p85 subunit [2]. The method can be used with different types of adherent cells, including GM7373 and human umbilical vascular endothelial cells (HUVECs), different stimuli [10] and following not only VEGFR2 but also other types of receptors (e.g. β3 integrin) [2]. As an alternative to fluorescent protein-tagged receptors, a 4'phosphopantetheinyl transferase (PPTase)-based labeling can be used to label S6-Tag receptors specifically exposed on membrane and visualize their membrane dynamics and eventually their internalization [12]. While in the previous studies we quantified only the amount of receptor reaching the substratum, here we describe the quantification of the distribution of the entire pool of receptors in the cell in time.

## Supporting information

**S1 File.**
(PDF)

## Acknowledgments

The authors performed experiments at the Imaging Platform at DMMT, University of Brescia.

## Author Contributions

**Conceptualization:** Ravelli Cosetta, Mitola Stefania.

**Data curation:** Ravelli Cosetta.

**Formal analysis:** Ravelli Cosetta.

**Funding acquisition:** Mitola Stefania.

**Investigation:** Ravelli Cosetta, Corsini Michela, Ventura Anna, Domenichini Mattia, Grillo Elisabetta, Mitola Stefania.

**Supervision:** Mitola Stefania.

**Writing – original draft:** Ravelli Cosetta, Corsini Michela.

**Writing – review & editing:** Ravelli Cosetta, Mitola Stefania.

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
