## [Decision Letter · Decision Letter 0]

3 Apr 2024

PONE-D-24-04439Alternative method to visualize receptor dynamics in cell membranes.PLOS ONE

Dear Dr. Mitola,

Thank you for submitting your manuscript to PLOS ONE. After careful consideration, we feel that it has merit but does not fully meet PLOS ONE’s publication criteria as it currently stands. Therefore, we invite you to submit a revised version of the manuscript that addresses the points raised during the review process.

We look forward to receiving your revised manuscript.

Kind regards,

Eleftherios Paschalis Ilios

Academic Editor

PLOS ONE

Journal Requirements:

2. Please include a link to the protocols.io entry in your Methods section.

3. We note you have not yet provided a protocols.io PDF version of your protocol and/or a protocols.io DOI. When you submit your revision, please provide a PDF version of your protocol as generated by protocols.io (the file will have the protocols.io logo in the upper right corner of the first page) as a Supporting Information file. The filename should be S1_file.pdf, and you should enter “S1 File” into the Description field. Any additional protocols should be numbered S2, S3, and so on. Please also follow the instructions for Supporting Information captions [https://journals.plos.org/plosone/s/supporting-information#loc-captions]. The title in the caption should read: “Step-by-step protocol, also available on protocols.io.”

Please assign your protocol a protocols.io DOI, if you have not already done so, and include the following line in the Materials and Methods section of your manuscript: “The protocol described in this peer-reviewed article is published on protocols.io (https://dx.doi.org/10.17504/protocols.io.[...]) and is included for printing purposes as S1 File.” You should also supply the DOI in the Protocols.io DOI field of the submission form when you submit your revision.

If you have not yet uploaded your protocol to protocols.io, you are invited to use the platform’s protocol entry service [https://www.protocols.io/we-enter-protocols] for doing so, at no charge. Through this service, the team at protocols.io will enter your protocol for you and format it in a way that takes advantage of the platform’s features. When submitting your protocol to the protocol entry service please include the customer code PLOS2022 in the Note field and indicate that your protocol is associated with a PLOS ONE Lab Protocol Submission. You should also include the title and manuscript number of your PLOS ONE submission.

6. Please ensure that you refer to Figure 1 in your text as, if accepted, production will need this reference to link the reader to the figure.

**Additional Editor Comments:**

The two reviewers made valid comments which need to be addressed before this paper is considered further. Please review the comments and address them to the best of your ability.

Thank you

Reviewers' comments:

Reviewer's Responses to Questions

**Comments to the Author**

1. Does the manuscript report a protocol which is of utility to the research community and adds value to the published literature?

Reviewer #1: Yes

Reviewer #2: Yes

2. Has the protocol been described in sufficient detail?

To answer this question, please click the link to protocols.io in the Materials and Methods section of the manuscript (if a link has been provided) or consult the step-by-step protocol in the Supporting Information files.

The step-by-step protocol should contain sufficient detail for another researcher to be able to reproduce all experiments and analyses.

Reviewer #1: Partly

Reviewer #2: Yes

3. Does the protocol describe a validated method?

Reviewer #1: Yes

Reviewer #2: Yes

4. If the manuscript contains new data, have the authors made this data fully available?

Reviewer #1: N/A

Reviewer #2: Yes

**5. Is the article presented in an intelligible fashion and written in standard English?**

Reviewer #1: Yes

Reviewer #2: Yes

6. Review Comments to the Author

Reviewer #1: The proposed manuscript by R. Cosetta and al. is dedicated to the development of a new approach to determine the receptor dynamics in the cell membrane. This approach was developed to use technology easily available in most research centers. This technological manuscript well describes the interest of such approach. However, some important points should be addressed to improve this manuscript:

- The authors indicated that this approach was "...an economical and simple assay to follow and characterize membrane receptor dynamics..." and that the use of epifluorescence microscopy (EM) was possible. However, in their descriptions, the authors used confocal microscopy and some related techniques such as image stacking (not really accessible in EM). The authors should describe the use of EM in this approach.

- The number of cells analyzed was not reported. The number of experiments was also not reported. These items should be reported.

- How was the percentage of positive areas determined? Image analysis ? Software used ? procedure ?

- The authors do not discuss other methods to study receptor dynamics. This should be discussed. The advantages and limitations of this new approach should also be discussed.

- The authors cited reference 17 for analysis of VEGFR and other receptor expression, but this reference did not use the same method. This should be discussed.

- The authors do not indicate whether the internalization of the receptors could be detected with this method ? or not.

Reviewer #2: The authors provide an alternative cost effective method for visualizing receptor dynamics in cell membranes.The protocol is well-described, providing clear step-by-step instructions for conducting the experiments. There are few suggestions that would help to improve the manuscript:

1. The introduction needs to be more general rather than focusing on one type of membrane receptors.

2. Figure 1: Experimental layout - the steps shall be described in details with graphical/pictorial representation of steps making it easier for the readers to follow.

3. Authors should incorporate the practical problems faced and troubleshooting they performed.

4. Lastly, it would be good to add a table showing the limitations and advantages of the existing techniques including the one described by the authors.

7. PLOS authors have the option to publish the peer review history of their article (what does this mean?). If published, this will include your full peer review and any attached files.

Reviewer #1: **Yes: **Alain COUVINEAU

Reviewer #2: No

---

## [Author Response · Author response to Decision Letter 0]

24 Apr 2024

We have addressed reviewers’ criticisms (in italics) as it follows:

1. Does the manuscript report a protocol which is of utility to the research community and adds value to the published literature?

Reviewer #1: Yes

Reviewer #2: Yes

We thank reviewers for appreciating the utility of our protocol and for recognizing its novelty.

2. Has the protocol been described in sufficient detail?

To answer this question, please click the link to protocols.io (DOI: dx.doi.org/10.17504/protocols.io.rm7vzj8z8lx1/v1 (Private link for reviewers: https://www.protocols.io/private/46322724FDB111EEB7ED0A58A9FEAC02 to be removed before publication.) in the Materials and Methods section of the manuscript or consult the step-by-step protocol in the Supporting Information files. The step-by-step protocol should contain sufficient detail for another researcher to be able to reproduce all experiments and analyses.

Reviewer #1: Partly

Reviewer #2: Yes

Following the Editor suggestion the step-by-step protocol, included in Supporting information in the previous version, was included and published into protocols.io . Please see: 

Please see DOI: dx.doi.org/10.17504/protocols.io.rm7vzj8z8lx1/v1 (Private link for reviewers: https://www.protocols.io/private/46322724FDB111EEB7ED0A58A9FEAC02 to be removed before publication.)

3. Does the protocol describe a validated method?

Reviewer #1: Yes

Reviewer #2: Yes

Here we described immobilized VEGF-A-recruited receptor to demonstrate the cogency of our method. The method allowed us to compare different experimental conditions such as different stimuli, inhibitors, receptors mutants and multiple receptors and co-receptors at the same time (see Damioli et al.).

4. If the manuscript contains new data, have the authors made this data fully available?

Reviewer #1: N/A

Reviewer #2: Yes

We added statistic detail s in figure captions.

5. Is the article presented in an intelligible fashion and written in standard English?

Reviewer #1: Yes

Reviewer #2: Yes

The typos and grammar errors were corrected. 

Reviewer Comments to the Author:

Reviewer #1

The proposed manuscript by R. Cosetta and al. is dedicated to the development of a new approach to determine the receptor dynamics in the cell membrane. This approach was developed to use technology easily available in most research centers. This technological manuscript well describes the interest of such approach.

However, some important points should be addressed to improve this manuscript:

- The authors indicated that this approach was "...an economical and simple assay to follow and characterize membrane receptor dynamics..." and that the use of epifluorescence microscopy (EM) was possible. However, in their descriptions, the authors used confocal microscopy and some related techniques such as image stacking (not really accessible in EM). The authors should describe the use of EM in this approach. 

Working in Z-stack and quantifying fluorescence in each slice, it is important to have a good 3D sectioning. We achieved it using structured illumination technology which is less expensive and time-consuming than a confocal laser microscopy. Alternatively a deconvolution algorithm ( plugins for Fiji or DeconvolutionLab and DeconvolutionLab2 (EPFL) (https://bigwww.epfl.ch/deconvolution/) can be used to analyze fluorescent stack image series . See lines 196-202 and Table 1

- The number of cells analyzed was not reported. The number of experiments was also not reported. These items should be reported. 

We apologize for forgetting to include the number of analyzed cells in each experiment and the number of experiments carried out. For each experiment, repeated three times, 20 cells were analyzed at each time point. We added this information in the Expected Results section (see linse 161-162). 

- How was the percentage of positive areas determined? Image analysis? Software used? procedure?

All the information regarding image quantification is reported in Supporting information: Step 5, Image quantification and data analysis (day 5). Briefly, for each experiment images were acquired with the same exposure time. Then, Z-stacks were analyzed with Fiji software. After defining a threshold, we drew a ROI for each cell in which we quantified the number of fluorescence-positive pixels in every slice of the Z-stack. Finally, we calculated the percentage of the receptor present in every slice to visualize its distribution.

- The authors do not discuss other methods to study receptor dynamics. This should be discussed. The advantages and limitations of this new approach should also be discussed.

We thank both Reviewers for the suggestion of deepening advantages and limitations of the methods currently available to study receptor dynamics on cell membranes. To address it, we decided to include a Table in Discussion section (Table 1, line 222) describing positive and negative aspects of the proposed protocol compared with the other two main methodologies: FRAP and SPT.

- The authors cited reference 17 for analysis of VEGFR and other receptor expression, but this reference did not use the same method. This should be discussed.

In the manuscript we refer to the papers by Damioli et al. (DOI) and Ravelli et al. (DOI) in which this protocol was used. In those publications, however, we focused our attention only on the quantification of receptors (VEGFR2 and Beta3 integrin) recruited in close contact with the immobilized factors. Here we analyzed the distribution, in time, of the entire pool of receptor.

- The authors do not indicate whether the internalization of the receptors could be detected with this method ? or not. 

We added a comment on the possibility of detection receptor internalization in lines 218-222.

Reviewer #2:

The authors provide an alternative cost effective method for visualizing receptor dynamics in cell membranes. The protocol is well-described, providing clear step-by-step instructions for conducting the experiments. There are few suggestions that would help to improve the manuscript:

1. The introduction needs to be more general rather than focusing on one type of membrane receptors. 

We introduced the different classes of membrane receptors in the introduction section ( see lines 32-36) , Then we focused only on one pass transmembrane receptors. Although we have no reason to think that the described protocol cannot be used to follow all kinds of receptors, we have alway applied it to analyze the dynamics of single-pass receptors ( e.i tyrosine kinase and integrins). 

2. Figure 1: Experimental layout - the steps shall be described in details with graphical/pictorial representation of steps making it easier for the readers to follow.

Following the reviewer’s suggestion we included in the revised version (Fig 2) a cartoon which recapitulates the steps from the coverslip flipping to image quantification. Fig 2: Cartoon of steps 3-5 of experimental flow chart (see lines 89, 147 and Fig2).

3. Authors should incorporate the practical problems faced and troubleshooting they performed.

The practical problem and the critical steps were included into DOI: dx.doi.org/10.17504/protocols.io.rm7vzj8z8lx1/v1 (Private link for reviewers: https://www.protocols.io/private/46322724FDB111EEB7ED0A58A9FEAC02 to be removed before publication.)

4. Lastly, it would be good to add a table showing the limitations and advantages of the existing techniques including the one described by the authors.

We included limitations and advantages in Table 1 (Advantages and disadvantages of microscopy techniques for the study of molecule relocation) and described in the Discussion section of the revised version. See Table 1, line 222 .

Sincerely 

Prof.ssa Stefania Mitola

---

## [Decision Letter · Decision Letter 1]

8 May 2024

Alternative method to visualize receptor dynamics in cell membranes.

PONE-D-24-04439R1

Dear Dr. Mitola,

We’re pleased to inform you that your manuscript has been judged scientifically suitable for publication and will be formally accepted for publication once it meets all outstanding technical requirements.

Kind regards,

Eleftherios Paschalis Ilios

Academic Editor

PLOS ONE

Additional Editor Comments (optional):

Thank you for addressing the comments

Reviewers' comments:

Reviewer's Responses to Questions

**Comments to the Author**

1. Does the manuscript report a protocol which is of utility to the research community and adds value to the published literature?

Reviewer #1: Yes

Reviewer #2: Yes

2. Has the protocol been described in sufficient detail?

To answer this question, please click the link to protocols.io in the Materials and Methods section of the manuscript (if a link has been provided) or consult the step-by-step protocol in the Supporting Information files.

The step-by-step protocol should contain sufficient detail for another researcher to be able to reproduce all experiments and analyses.

Reviewer #1: Yes

Reviewer #2: Yes

3. Does the protocol describe a validated method?

Reviewer #1: Yes

Reviewer #2: Yes

4. If the manuscript contains new data, have the authors made this data fully available?

Reviewer #1: Yes

Reviewer #2: Yes

**5. Is the article presented in an intelligible fashion and written in standard English?**

Reviewer #1: Yes

Reviewer #2: Yes

6. Review Comments to the Author

Reviewer #1: The proposed manuscript by R. Cosetta and al. is dedicated to the development of a

new approach to determine the receptor dynamics in the cell membrane. This

approach was developed to use technology easily available in most research centers.

This technological manuscript well describes the interest of such approach.

The authors having totally replied to my previous comments. In this context, the present manuscript should be accepter for publication.

Reviewer #2: The authors have successfullly addressed all the comments and revised the manuscript accordingly.

7. PLOS authors have the option to publish the peer review history of their article (what does this mean?). If published, this will include your full peer review and any attached files.

Reviewer #1: **Yes: **Alain Couvineau

Reviewer #2: **Yes: **Jyoti Sharma

---

## [Editor Report · Acceptance letter]

31 May 2024

PONE-D-24-04439R1 

PLOS ONE

Dear Dr. Stefania, 

I'm pleased to inform you that your manuscript has been deemed suitable for publication in PLOS ONE. Congratulations! Your manuscript is now being handed over to our production team.

Kind regards, 

on behalf of

Dr. Eleftherios Paschalis Ilios 

Academic Editor

PLOS ONE